# Effect of Fibroblast Growth Factor 10 and an Interacting Non-Coding RNA on Secondary Hair Follicle Dermal Papilla Cells in Cashmere Goats’ Follicle Development Assessed by Whole-Transcriptome Sequencing Technology

**DOI:** 10.3390/ani13132234

**Published:** 2023-07-07

**Authors:** Yuan Gao, Weiguo Song, Fei Hao, Lei Duo, Xiaoshu Zhe, Chunyan Gao, Xudong Guo, Dongjun Liu

**Affiliations:** State Key Laboratory of Reproductive Regulation and Breeding of Grassland Livestock, School of Life Sciences, Inner Mongolia University, Hohhot 010070, China; 21908009@mail.imu.edu.cn (Y.G.); 32008255@mail.imu.edu.cn (W.S.); feihao@imu.edu.cn (F.H.); 21908010@mail.imu.edu.cn (L.D.); 22208021@mail.imu.edu.cn (X.Z.); 32208063@mail.imu.edu.cn (C.G.); bixdguo@imu.edu.cn (X.G.)

**Keywords:** cashmere goats, hair follicle development, FGF10

## Abstract

**Simple Summary:**

Cashmere goats are one of the most important animal breeds in northern China. Cashmere is characterised by good quality and high yield and represents an excellent textile material. However, the molecular mechanism underlying the development of cashmere hair follicles is poorly understood, which limits the further development of the cashmere industry. Here, we used whole-transcriptome sequencing to analyse goat embryonic skin samples and carried out experimental verification. We found that fibroblast growth factor 10 and the interacting non-coding RNA miR-184 play important roles in dermal papilla cells of cashmere goat secondary hair follicles. This discovery not only further improves the theoretical understanding of the development of cashmere wool follicles but also has significant implications for functional research on fibroblast growth factor 10 and miR-184.

**Abstract:**

Cashmere, a keratinised product of secondary hair follicles (SHFs) in cashmere goats, holds an important place in international high-end textiles. However, research on the complex molecular and signal regulation during the development and growth of hair follicles (HFs), which is essential for the development of the cashmere industry, is limited. Moreover, increasing evidence indicates that non-coding RNAs (ncRNAs) participate in HF development. Herein, we systematically investigated a competing endogenous RNA (ceRNA) regulatory network mediated by circular RNAs (circRNAs), microRNAs (miRNAs), and messenger RNAs (mRNAs) in skin samples of cashmere goat embryos, using whole-transcriptome sequencing technology. We obtained 6468, 394, and 239 significantly differentially expressed mRNAs, circRNAs, and miRNAs, respectively. These identified RNAs were further used to construct a ceRNA regulatory network, mediated by circRNAs, for cashmere goats at a late stage of HF development. Among the molecular species identified, miR-184 and fibroblast growth factor (FGF) 10 exhibited competitive targeted interactions. In secondary HF dermal papilla cells (SHF-DPCs), miR-184 promotes proliferation, inhibits apoptosis, and alters the cell cycle via the competitive release of FGF10. This study reports that FGF10 and its interaction with ncRNAs significantly affect SHF-DPCs, providing a reference for research on the biology of HFs in cashmere goats and other mammals.

## 1. Introduction

Cashmere, a keratinised product of secondary hair follicles (SHFs) in cashmere goats, is characterised by its softness, lightness, fineness, and heat retention, and holds an important place in international high-end textiles. Hair follicles (HF) are one of the most important organs in the skin, generating and supporting hair growth as well as providing skin protective and sensory functions. HF are subject to complex molecular and signal regulation during development and growth; however, the underlying mechanisms are not fully understood. Therefore, research on HF developmental mechanisms is important. During HF growth, signalling elements and key molecules regulate different stages of HF development. Some signalling pathways, such as the Wnt, BMP, MAPK, and PI3K-AKT pathways, have been widely studied [1,2,3]. For example, activation of the MAPK signalling pathway can promote self-renewal and proliferation of HF stem cells, thereby maintaining HF development and growth [4]. In diseases, including hair loss and alopecia areata, abnormal activation of the MAPK signalling pathway can lead to apoptosis and aberrant differentiation of HF cells, leading to atrophy and loss of HFs [5]. In terms of key molecules, non-coding RNAs (ncRNAs) play important biological roles. For example, long non-coding (lnc) RNA-PCAT1 in dermal papilla cells (DPCs) can adsorb microRNA (miR)-329 and promote HF regeneration, which has the potential to treat hair loss [6]. Similarly, miR-203 can participate in HF development by competing with multiple targeted messenger RNAs (mRNAs) for competing endogenous RNA (ceRNA) [7]. Therefore, an in-depth understanding of the molecular mechanisms underlying HF development may help improve the yield and quality of cashmere and provide a scientific basis for the treatment of HF-related diseases.

For cashmere goats, the formation of HF can be traced back to early embryonic development. According to morphological changes, this process can be roughly divided into three stages: induction, organ formation, and differentiation. According to the occurrence time and structural characteristics, follicles are divided into primary and secondary HFs (PHF and SHF, respectively). During the E60 period, the dermal papilla cell population begins to form in the surface layer of the skin, forming initial cell clusters. During the E75 period, as the embryo develops, the HF primordia begin to expand towards the lower layer of the skin. This expansion leads to the formation of the HF epidermis and papilla, forming the primitive PHFs. During the E85 period, the development of SHFs commences, with a small proportion of SHFs emerging adjacent to the root of a portion of PHF and laterally. During the E125 period, notable events occur in the differentiation of HFs, including the formation of the inner root sheath, sebaceous gland, hair erector muscle, and other structures; the protrusion of the hair trunk from the body surface; the formation of the initial trichome structure; and the basic completion of HF growth and development processes [8]. In addition, the development of cashmere follicles is regulated by various factors, including proteins, growth factors, and signalling pathways. Research on the occurrence and development of cashmere follicles has shown that signalling pathways, such as the Wnt and BMP pathways, play an important role in the formation of HFs [9]. In addition, many lncRNAs and miRNAs are also involved in regulating the development of cashmere follicles [10,11]. The HF cycle of cashmere goats includes three stages; anagen, catagen, and telogen are involved, which collectively govern the growth rate, length, and quality of cashmere hair [12,13]. Therefore, investigating the occurrence, development, and periodic growth of HFs in cashmere goats not only helps to understand the regulatory mechanisms of wool fibre yield and quality but also provides an important theoretical basis for the cashmere goat breeding industry.

High-throughput sequencing technology, also known as next-generation sequencing or large-scale parallel sequencing, enables the simultaneous sequencing of hundreds of thousands to millions of nucleic acid molecules [14]. Transcriptomic analysis of cashmere goats under different photoperiods revealed differentially expressed miRNAs and mRNAs affecting the growth of cashmere [15]. In addition, high-throughput sequencing studies have enabled the identification of cashmere shape characteristics in cashmere goats [16], the identification of key genes and proteins of different cashmere types [17,18,19,20], and the analysis of goats’ milk production traits [21]. This technology can comprehensively reveal the molecular regulatory network and mechanism of cashmere shape in cashmere goats from a macroscopic and multidimensional perspective, providing an important theoretical basis for in-depth research on cashmere goats.

The present study aimed to systematically investigate a ceRNA regulatory network mediated by circular RNAs (circRNAs), miRNAs, and mRNAs, using skin tissues from Inner Mongolian cashmere goats at foetus stages (75 and 125 days). After quality control, sequence alignment, transcript assembly, ncRNA identification and prediction, and differential expression (DE) analysis, differentially expressed ncRNAs and mRNAs (DEncRNAs and DEmRNAs, respectively) were screened, and candidate genes for HF development were screened using Gene Ontology (GO) and Kyoto Encyclopedia of Genes and Genomes (KEGG) enrichment analysis. The targeting relationship was verified using a dual-luciferase reporter system, and the effects on SHF-DPC proliferation, apoptosis, and cell cycle were explored. The results of this study will provide a database for in-depth research on the developmental mechanisms of HFs in cashmere goats, providing theoretical support for the refinement of HF biology in cashmere goats and other mammalian species. Additionally, our study provides preliminary theoretical support for the development of new genetic breeding materials for Inner Mongolia cashmere goats with excellent cashmere traits and the innovative use of the core germplasm resources of cashmere goats.

## 2. Materials and Methods

### 2.1. Sample Preparation

Two months before the onset of oestrus, female Albas goats with similar cashmere production performance were selected to form a breeding herd, and semen from one ram was collected and inseminated with a date of embryonic development day 0 (E0) after semen quality checks. Pregnant ewes were regrouped and allowed to graze for 12–14 h per day. During this period, they were allowed to drink water three to four times. The daily supplementary diet was 150 g each, consisting of mainly maize, soya, and cereals, with 1.5% salt, 1.0% trace elements, and 0.5% vitamins. At E75 and E125, 6 cashmere goat foetuses were obtained using surgical methods from 6 ewes (3 foetuses were obtained from E75 and 3 foetuses were obtained from E125). All skin tissue was collected, and the skin from the lateral part of the foetal body was used for morphological observation and RNA sequencing in this study.

### 2.2. Library Preparation and Illumina Hiseq xten/NovaSeq6000 Sequencing

Total RNA was extracted from tissues using TRIzol reagent (Sigma-Aldrich, St Louis, MO, USA) according to the manufacturer’s instructions, and RNA quality was evaluated after removing genomic DNA. Only high-quality RNA samples (optical density (OD)_260_/OD_280_ 1.8–2.2, OD_260_/OD_230_ ≥ 2.0, RNA integrity number (RIN) ≥ 8, 28S:18S ≥ 1.0, >10 μg) were used to build sequencing libraries

Total RNA (5 µg) was prepared as an RNA-seq library using the TruSeq Stranded Total RNA Kit from Illumina (San Diego, CA, USA). Ribosomal RNA was removed using the Ribo-Zero, and then the first cDNA strand was synthesised using random hexamer primers. AMPure XP beads were used to separate the double-stranded (ds) cDNA from the second-strand reaction mix. Finally, multiple indexing adapters were ligated to the ends of the ds cDNA. PCR was performed using Phusion DNA polymerase (New England Biolabs, Ipswich, MA, USA) to amplify a 200–300 bp cDNA target fragment to construct a library. After quantification using TBS380, the paired-end RNA-seq library was sequenced using Illumina HiSeq xten/NovaSeq6000 (2 × 150 bp read length).

In addition, 3 μg of total RNA was ligated with sequencing adapters using the TruSeq Small RNA Sample Prep Kit (Illumina). Subsequently, cDNA was synthesised by reverse transcription and amplified with 12 PCR cycles to produce libraries. After quantification using TBS380, deep sequencing was performed by Shanghai Majorbio Bio-Pharm Biotechnology Co., Ltd. (Shanghai, China).

### 2.3. Sequencing Data Quality Control

The Illumina sequencing platform converts sequencing image signal recognition into a Fastq format with storage as raw data. Illumina sequencing generates hundreds of millions of readings per run, and a large amount of data cannot show the quality of specific individual reads. The construction of a suitable sample library can only be determined through the application of classification statistics and analysis. The base quality, error rate, and content distribution of each sample are important reference indicators.

SeqPrep (https://github.com/jstjohn/SeqPrep (accessed on 8 July 2021)) and Sickle (https://github.com/najoshi/sickle (accessed on 8 July 2021)) software were used to remove any repetitive splice sequences, uncertain base information, low-quality sequences, and sequences that were too long or too short from the raw data, as low-quality reads can affect the accuracy of the sequencing results.

### 2.4. Sequence Alignment

Raw data after quality control were compared with the reference genome of goats (*Capra hircus*), using TopHat.v2.1.1 (http://ccb.jhu.edu/software/tophat/index.shtml (accessed on 8 July 2021)) [22] and HISAT2.2.1.0 (http://ccb.jhu.edu/software/hisat2/index.shtml (accessed on 8 July 2021)) [23] software. The reference genome version used was ARS1 (http://asia.ensembl.org/Capra_hircus/Info/Index (accessed on 8 July 2021)). The final comparison results, after quality evaluation, were used as mapped reads for subsequent analysis.

### 2.5. Transcript Assembly and Functional Annotation

Based on the reference genome, StringTie.1.3.3b (https://ccb.jhu.edu/software/stringtie/ (accessed on 8 July 2021)) software was used to assemble and splice-map reads, compare them to known transcripts, obtain transcripts without annotation information, and perform functional annotation of potential new transcripts.

### 2.6. Identification and Prediction of ncRNAs

The identification of circRNAs was performed using CIRI2 (https://sourceforge.net/projects/ciri/ (accessed on 9 July 2021)) to identify reads that had not been compared to the reference genome. The Circle software was used to predict circRNAs and compare overlaps with the circBase database (http://circbase.org/ (accessed on 9 July 2021)) for analysis.

Identification of miRNAs involved comparing the reads of the reference genome with the miRBase (https://www.mirbase.org/ (accessed on 9 July 2021)) and Rfam (https://rfam.org/ (accessed on 9 July 2021)) databases to obtain known miRNA and ncRNA annotation information and using the miRDeep2 (mirdeep2-documentation) software to predict the secondary structure of reads without annotation. Based on the predicted results, features—including energy values and ribonuclease site information—were used to filter and identify new miRNAs.

### 2.7. Differential Expression Analysis

RSEM.1.3.1 (http://deweylab.biostat.wisc.edu/rsem/ (accessed on 9 July 2021)) software was used to analyse expression levels for the assembled transcripts, and the relative expression was quantified as transcripts per million reads (TPM). DESeq2.1.10.1 (http://bioconductor.org/packages/stats/bioc/DESeq2/ (accessed on 9 July 2021)) software was used for differential analysis of expression level, with a default *p*-value < 0.05 and fold differences of two. Multiple tests and corrections using the Benjamini–Hochberg (BH) method were used to improve the accuracy of the analysis.

### 2.8. ncRNA Target Gene Prediction

Based on the principle that animal miRNA relies on tight binding of the 5′-terminal 2–8 nucleotides of the seed sequence to the 3′ untranslated region (UTR) of the target gene, DEmiRNA target genes were obtained using miRanda.0.10.80 (https://www.miranda.org/ (accessed on 9 July 2021)) and RNAhybrid.2.1 (https://bibiserv.cebitec.uni-bielefeld.de/rnahybrid/ (accessed on 9 July 2021)) software.

### 2.9. Functional Enrichment Analysis

GO enrichment analysis was performed using Goatools.0.6.5 software and KEGG enrichment analysis was performed using the R package. Analytical results were corrected using multiple BH tests, and *p*-values < 0.05 after correction were considered to represent significant enrichment.

### 2.10. Construction of a ceRNA Network

DEncRNAs obtained through differential expression analysis were assessed using the ceRNA theory. After interaction analysis and negative correlation filtering for miRNA and candidate ceRNA expression levels, a Pearson correlation coefficient < 0 and *p* < 0.05 were used to construct and visualise a ceRNA regulatory network using Cytoscape 3.9.1.

### 2.11. Construction of Vectors

To construct an overexpression vector for FGF10, the complete coding sequence of the goat FGF10 gene (GeneID: 102174339) was obtained from the NCBI database. Specific primers were designed using SnapGene.4.1.9 software, and BamHI and EcoRI were used as restriction endonucleases to connect to the pcDNA 3.1 vector (RRID: Addgene-24386).

FGF10 siRNA was synthesised from the Gemma Gene Co. (Suzhou, China), and three different sites were selected for interference.

To achieve overexpression and inhibition of miR-184 in cells, the corresponding mimics and inhibitors, as well as the corresponding non-coding mimics and inhibitors (synthesised by the Gemma Gene Co.), were synthesised using chemical synthesis (Appendix A).

### 2.12. Morphological Observation

Foetal skin samples (1 cm^2^) were cut from the side of the body at the E75 and E125 time points, rinsed with phosphate-buffered saline (PBS), and immediately fixed using 4% paraformaldehyde for 24 h. After gradient dehydration, alcohol–benzene replacement, paraffin immersion, sectioning, haematoxylin–eosin staining, and microscopic observations were performed.

### 2.13. Target Verification

Based on the predicted binding sites, wild-type (WT) and mutant (MUT) double-luciferase reporter gene vectors were constructed. Primers containing XhoI (5′-C^TCGAG-3′) and XbaI (5′-T^CTAGA-3′) restriction endonuclease sites were designed on both sides of the binding site. Single-stranded fragments were synthesised by Huada Gene Co. (Beijing, China) and annealed to form double-stranded fragments (Appendix A). The linear double-luciferase reporter gene vector PmirGLO Vector was recovered, purified, and ligated with the annealed double-stranded fragment.

### 2.14. Assessment of Cell Proliferation

A total of 1 × 10^3^ cells were inoculated into 4 cell culture dishes and placed in a cell culture incubator at 37 °C in 5% CO_2_. Cell proliferation was assessed by measuring the optical density after adding 10 μL of a cell-counting kit 8 (Abbkine Scientific Co., Beijing, China) reagent at 12, 24, 36, and 48 h.

### 2.15. Detection of Apoptosis

Double staining of SHF-DPCs was carried out 48 h after transfection, and the culture solution was aspirated into a 1.5 mL centrifuge tube. Subsequently, cells digested with trypsin were collected and centrifuged at 2000 rpm for 5 min, and the supernatant was discarded. Cells were gently re-suspended in 400 μL 1× Binding Buffer, then 5 μL of Annexin V-FITC reagent (7sea Biotech Co., Shanghai, China) was added, and samples were mixed gently and incubated at room temperature in the dark for 15 min. Propidium Iodide (PI) reagent (10 μL) was added and gently mixed well, followed by incubation at a low temperature in the dark for 5 min. To avoid quenching of fluorescent substances, rapid-flow cytometry detection (Beckman Coulter, Brea, CA, USA) was performed.

### 2.16. Cell Cycle Analysis

A suspension of SHF-DPCs transfected for 48 h was collected into a 1.5 mL centrifuge tube and centrifuged at 2000 rpm for 5 min, and the supernatant was discarded. Pre-cooled PBS (1 mL) was used to slowly resuspend cells, which were then centrifuged at 2000 rpm for 5 min, and the supernatant was discarded. A small amount of pre-cooled 70% aqueous ethanol was added to the centrifuge tube and used to resuspend cells, which were then incubated overnight at −20 °C. After resuspension, 500 μL of the mixture (1 mL staining buffer + 25 μL PI + 20 μL RNase A) was added and incubated for 30 min, and the flow cytometry assay was completed within 5 h.

### 2.17. Reverse-Transcription Quantitative Polymerase Chain Reaction (RT-qPCR) Validation

Total RNA was reverse-transcribed to cDNA and RT-qPCR was performed using the GoScript reverse-transcription system (Promega, Madison, WI, USA) and a C1000 thermal cycler (Bio-Rad Laboratories, Hercules, CA, USA). β-Actin and U6 were used as internal controls. The 2^−ΔΔCT^ method was used to calculate relative gene expression.

### 2.18. Western Blotting

After sample processing, total protein was quantified using a bicinchoninic reagent kit (Thermo Fisher Scientific, Waltham, MA, USA). Samples were loaded at 30 μg protein volume with a 5 μL of a protein marker, and the protein concentration condition was set at 90 V for 30 min and the protein separation condition at 120 V for 50 min, depending on the protein size. The membrane was then transferred, washed with TBST (50 mL TBS + 450 mL ultrapure water + 250 μL Tween 20), and incubated with an FGF10 antibody (ab71794, RRID: AB_2049648, Abcam, Cambridge, UK) overnight at 4 °C. After washing with TBST, the membranes were then incubated with the secondary antibody, washed for 1 h, and then visualised for protein quantification.

## 3. Results

### 3.1. HF Morphogenesis

Skin samples obtained from cashmere goat foetuses were subjected to haematoxylin and eosin staining at E75 and E125. Certain PHFs had developed at E75, but a complete HF structure had not yet formed. SHFs are absent during the initial phase of HF development. HF development enters a stage of rapid differentiation at E125. The primary HF morphology tended to be complete at this time; partial PHFs had matured and extended out of the body surface, and a clear HF structure was visible (Figure 1). SHFs develop and differentiate rapidly and are distributed around PHFs. Therefore, the results confirm the occurrence and development of HFs in cashmere goats. Samples from the 75- and 125-day time points (encompassing the initiation and rapid differentiation stages of HF development) were considerably different, indicating that sampling was successfully representative of the developmental process.

### 3.2. RNA Sequencing and cDNA Library Construction

In this study, we aimed to obtain a global view of the skin transcriptome (including mRNAs, miRNAs, and circRNAs) at embryonic stages E75 and E125 in cashmere goats and determine which RNAs are involved in HF development. Six samples were sequenced and cDNA libraries were constructed (long RNA-seq and small RNA-seq). A total of 813,141,270 reads were obtained from the long RNA-seq libraries and 807,089,298 clean reads were obtained from the raw data after stringent quality control. The percentage of clean reads among raw reads was >99% for each library. The Q30 values were above 95%. In total, 91,915,124 reads were obtained from the small RNA-seq libraries and 89,045,016 clean reads were obtained from the raw data after stringent quality control. The percentage of clean reads among raw reads ranged from 96.1–97.7% in the different libraries. Q30 were above 95%. Six samples from the long RNA-seq and sRNA-seq libraries that were individually aligned to the reference genome showed alignment efficiencies ranging from 97.72–97.99% and 82.95–94.16%, respectively (Appendix A). Inter-sample correlation analysis and principal component analysis were carried out based on sample expression; the biological replicates for each sample were in good agreement, and discrimination of sample groups at different time points was clear (Appendix A). The results confirm high-quality library construction and sequencing and the good biological reproducibility of samples for subsequent biological analysis.

### 3.3. Expression Analysis and Functional Annotation of mRNAs

mRNA expression levels were quantified using the quantitative metric TPM in the long RNA-seq libraries. In total, 22,108 mRNAs were identified in the E75 group, including 20,632 known and 1476 novel mRNAs. In total, 22,154 mRNAs were identified in the E125 group, including 20,672 known and 1482 novel mRNAs. Violin plots were used to visualise gene expression in each sample (Figure 2a). The data were optimised, and the expression of 19,744 mRNAs was found to be significantly higher than that of the remaining mRNAs. A Venn diagram revealed that 16,627 mRNAs were expressed in both groups, with 1590 mRNAs specifically expressed in the E75 group and 1577 mRNAs specifically expressed in the E125 group (Figure 2b). In total, 6468 DEmRNAs were identified in both groups, including 3234 upregulated and 3234 downregulated mRNAs. DEmRNAs were visualised (Figure 2c), and mRNAs with significant differences were selected to draw a cluster analysis heatmap (Figure 2d).

GO annotation was carried out to explore the functions of the DEmRNAs (Figure 2e), which indicated that DEmRNAs were mainly enriched in cell progression in biological processes, cell components, and the binding role in molecular functions. Subsequently, the biological functions of the DEmRNAs were subjected to GO enrichment analysis (Figure 2f). Significant enrichment was observed in several biological pathways, including HF morphogenesis (GO:0031069), hair cycle processes (GO:0022405), epidermal cell differentiation (GO:0009913), and positive regulation of animal organ morphogenesis (GO:0110110). KEGG enrichment analysis of the DEmRNAs indicated significant enrichments in the Ras, MAPK, and Rap1 signalling pathways, which are important for HF development (Figure 2g). Keratin (KRT), keratin-associated protein (KRTAP), oestrogen signalling pathway, and human papillomavirus infection signalling pathways were also significantly enriched in the *Staphylococcus aureus* infection signalling pathway. This result indicates that signalling pathways related to HF development are highly activated, and the KRTs and KRTAP, which are involved in HF formation, are also strongly expressed during SHF development in the skin at the embryonic stage in cashmere goats.

### 3.4. Expression Analysis and Functional Annotation of circRNAs

Analysis of circRNA data predicted that 36,259 circRNAs were present without chromosomal distribution bias (Figure 3a), with 67.89%, 18.48%, and 13.63% belonging to exon, intron, and intergenic regional patterns (Figure 3b), respectively. Length classification statistics for circRNAs (Figure 3c) showed that the largest group had sequence lengths under 5 kb. However, most of these potential circRNAs had an expression that was too low for subsequent analysis. Data optimisation showed that there were 698 circRNAs with real expression levels in the E75 libraries and 668 potential circRNAs in the E125 libraries. Among these, 457 circRNAs were expressed in both groups, 241 were specifically expressed in the E125 group, and 211 were specifically expressed in the E75 group (Figure 3d). In total, 394 DEcircRNAs were identified in the E75 and E125 groups, including 154 upregulated and 240 downregulated circRNAs. A volcano plot and cluster analysis heatmap were constructed (Figure 3e and Figure 3f, respectively). The significant changes in the expression of circRNAs during this period suggest their potential involvement in SHF development.

### 3.5. Expression Analysis and Functional Annotation of miRNAs

The Capra_hircus smallRNA Seq libraries yielded an average of 803 mature miRNAs, including 418 known miRNAs and 385 novel predicted miRNAs, from the E75 group libraries. The three E125 libraries yielded an average of 791 mature miRNAs, of which 421 were known and 370 were newly predicted. Violin plots were used to represent miRNA expression in each sample (Figure 4a). A Venn diagram for the groups was prepared using miRNAs with expression levels > 1 in the E75 and E125 groups (Figure 4b). Of these, 445 miRNAs were expressed in both the E75 and E125 groups, 29 were specifically expressed in the E75 group, and 38 were specifically expressed in the E125 group. In total, 239 DEmiRNAs, including 97 upregulated and 142 downregulated miRNAs, were identified between the two groups (*p* < 0.05, fold difference 2). A volcano plot and cluster analysis heatmap were prepared for the DEmiRNAs (Figure 4c,d, respectively).

The target genes of the DEmiRNAs were predicted in the DEmRNA and DEcircRNA databases using miRanda software. Of these, 4612 were mRNAs and 323 were circRNAs. In addition, many interactions among miRNAs, ncRNAs, and mRNAs were identified for the development of HF in cashmere goats. For example, miRNA-214 was predicted to interact with 199 mRNAs and 42 circRNAs, whereas mir-195-5p was predicted to interact with 30 circRNAs and 107 mRNAs. In summary, there appears to be a complex set of miRNA–mRNA interactions during SHF development in cashmere goats that may be regulated by changing the expression profile of the target genes.

The host gene sets (mRNAs) of the DEmiRNAs were then subjected to functional annotation and enrichment analysis using GO and KEGG (Figure 4e,f). GO results showed that the host genes of DEmiRNAs were enriched in negative regulation of cell projection organisation (GO:0031345), developmental growth involved in morphogenesis (GO:0060560), regulation of synapse assembly (GO:0051963), HF morphogenesis (GO:0031069), and the hair cycle processes (GO:0022405). The KEGG results for the host gene sets of DEmiRNAs were largely consistent with the enrichment results for the gene sets of DEmRNAs; any signalling pathways with important roles in HF development were similarly activated. These results show that miRNAs are involved in the development of HF by regulating the expression of host genes.

### 3.6. IPath Metabolic Pathway Analysis

DEmRNAs were compared with the host gene sets of DEmiRNAs for iPath metabolic pathway analysis. The host gene sets of DEmRNAs and DEmiRNAs were significantly enriched in glycosaminoglycan biosynthesis, glycosaminoglycan degradation, and some xenobiotic biodegradation and metabolism pathways for chondroitin sulphate and dermatan sulphate (Appendix A). DEmRNAs were also enriched in amino acid and energy metabolism pathways. The host genes of DEmiRNAs were most significantly altered in sugar biosynthesis and metabolic pathways in the skin and HF development processes of cashmere goat embryos.

### 3.7. Construction of ceRNA Network

The enrichment results showed that the target genes of both DEncRNAs and DEmRNAs were significantly enriched in HF regulation. Based on the positive correlation between DEcircRNA and DEmRNA expression, we constructed the ceRNA regulatory network with DEmiRNA as the core, DEcircRNA as the upstream region, and DEmRNA as the downstream region (Appendix A). The network contained 5581 ncRNA relationship pairs, covering 219 DEmiRNAs, 2477 DEmRNAs, and 232 DEcircRNAs. Owing to the size and complexity of the ceRNA network, we chose three species (miR-184, miR-200b, and miR-214-3p) for visualisation using Cytoscape 3.9.1 (Figure 5a). The ncRNA action axis predicted that chi-circRNA-0001141 competitively adsorbed miR-184 to release FGF10 (Figure 5b).

### 3.8. RT-qPCR Validation

To verify the RNA-seq results, we randomly selected four miRNAs, four circRNAs, and four mRNAs and assessed their expression levels using RT-qPCR (Appendix A). The correlation coefficients between the obtained results and the RNA-seq data were all greater than 0.8, demonstrating high reliability for the RNA-seq data (Appendix A).

### 3.9. Verification of the Targeting Relationship between FGF10 and ncRNA

Further analysis revealed that several HF functional genes were enriched in pathways known to be involved in HF development, with the FGF family and their receptors being particularly prominent. To further understand the role of the FGF family in HF development, we generated a collection of transcripts of the differential FGF family and its receptors. Enrichment analysis of the Sankey bubble map showed that the FGF family was also enriched in the MAPK, PI3K-AKT, RAS, and RAP1 pathways, as well as other pathways related to HF development and regeneration, suggesting that the FGF family and its receptors are involved in the regulation of HF processes (Appendix A). A temporal analysis of the pooled transcripts of the differential FGF family and its receptors revealed significant enrichment in three expression patterns (*p* < 0.05) (Appendix A). Among these, FGF10 expression trends were highly variable and differed significantly in expression at E75 and E125, suggesting its potential as a candidate gene for follow-up studies (Appendix A).

By analysing the sequence information of chi-circRNA-0001141, miR-184, and FGF10-3′-UTR, the binding sites of miR-184 and FGF10—as well as the binding sites of chi-circRNA-0001141 and miR-184—were predicted (Figure 6a,c). WT and MUT double-luciferase reporter gene recombinant plasmids were co-transfected with miR-184 mimics and mimics-NC into 293T and GFF cells according to the predetermined experimental grouping, and the fluorescence values of Firefly and Renilla in the corresponding groupings were detected (Figure 6b). In 293T and GFF cells, the fluorescence ratio of Firefly and Renilla cells was significantly lower in cells co-transfected with miR-184 mimics and Pmir-GLO-circRNA-WT than in control cells (*p* < 0.001). However, after co-transfection with miR-184 mimics and Pmir-GLO-circRNA-MUT, the fluorescence ratio recovered, indicating a targeted relationship between chi-circRNA-0001141 and miR-184.

In 293T and GFF cells, on co-transfection with miR-184 mimics and Pmir-GLO-FGFF10-WT-1, the fluorescence ratio of Firefly and Renilla was significantly lower than in control cells (*p* < 0.001). However, after co-transfection with miR-184 mimics and Pmir-GLO-FGFF10-MUT-1, the fluorescence ratio recovered; when miR-184 mimics and Pmir-GLO-FGF10-WT-2 were co-transfected, the fluorescence ratios of Firefly and Renilla were not significantly different from those of the control group, and the fluorescence ratios of the mutant group were not significantly different from those of the control group. The results demonstrate that miR-184 binds to FGF10 3′-UTR at the binding site predicted by miRanda (Figure 6d).

To further validate the targeting relationship, overexpression and interference of miR-184 were used to detect the expression of FGF10 at the transcriptional and protein levels, respectively. The transcriptional and protein expression levels of FGF10 were significantly lower in the miR-184 overexpression group than that in the control group (*p* < 0.01 and *p* < 0.001, respectively), indicating that miR-184 overexpression had an inhibitory effect on FGF10 expression (Figure 6e,f). Moreover, after interference with miR-184, both the transcriptional and protein expression levels of FGF10 were significantly higher compared to those in the control group (*p* < 0.01 and *p* < 0.001, respectively), indicating that interference with miR-184 promoted FGF10 expression (Figure 6g,h). These results indicate that changes in the expression level of miR-184 in SHF-DPCs can lead to changes in the expression level of FGF10, confirming a targeted relationship between miR-184 and FGF10.

### 3.10. miR-184 Inhibits the Proliferation of SHF-DPCs by Targeting FGF10

After 48 h of continuous monitoring of cells overexpressing or subject to interference, we observed that the OD value was significantly higher for the FGF10 overexpression group than for the control group at 36 h (*p* < 0.01) or 48 h (*p* < 0.05), whereas that of the FGF10 interference group was significantly lower than that of the control group at 36 or 48 h (*p* < 0.01). Moreover, the OD value of the FGF10 overexpression group was higher than that of the interference group, indicating that FGF10 overexpression promoted the proliferation of SHF-DPCs and that interference with FGF10 inhibits proliferation (Figure 7a). The OD value was significantly lower for the miR-184 overexpression group than for the control group at 36 h (*p* < 0.05) and 48 h (*p* < 0.01). The OD value of the miR-184 interference group was significantly higher than that of the control group at 36 h (*p* < 0.001) and 48 h (*p* < 0.05). Moreover, the OD value for the miR-184 interference group was higher than that for the overexpression group, indicating that miR-184 overexpression inhibited the proliferation of SHF-DPCs, whereas interference with miR-184 promoted it (Figure 7b).

### 3.11. miR-184 Inhibits Apoptosis in SHF-DPCs by Targeting FGF10

Using Annexin V-FITC and PI staining, flow cytometry was used to detect living, early apoptotic, late apoptotic, and necrotic cells with different expression levels of FGF10 and miR-184. The results of apoptosis detection showed that, 48 h after transfection, the apoptosis rate in the FGF10 overexpression group was significantly lower than that in the control group (*p* < 0.05), whereas the apoptosis rate in the FGF10 knockdown group was significantly higher than that in the control group (*p* < 0.01), indicating that FGF10 knockdown promotes apoptosis of SHF-DPCs and increases the number of apoptotic cells, whereas its overexpression inhibits apoptosis and reduces the number of apoptotic cells (Figure 8a,b). Moreover, 48 h after transfection, the apoptosis rate in the miR-184 overexpression group was significantly higher (*p* < 0.01)—whereas that of the miR-184 interference group was significantly lower (*p* < 0.001)—than that of the control group, indicating that miR-184 overexpression promoted the apoptosis of SHF-DPCs and increased the number of apoptotic cells, whereas interference with miR-184 inhibited this process and reduced the number of apoptotic cells (Figure 8c,d).

### 3.12. miR-184 Affects Cell Cycle Progression in SHF-DPCs by Targeting FGF10

Flow cytometry analysis of SHF-DPCs stained with PI revealed the effects of FGF10 and miR-184 on the cell cycle. Compared to results in the control group, the proportion of cells in the G2/M phase in the FGF10 overexpression group showed no significant difference, while that in the G0/G1 phase was significantly decreased (*p* < 0.001) and that in the S phase was significantly increased (*p* < 0.001) (Figure 9a). In the FGF10 interference group, the proportion of cells in the G0/G1 phase increased significantly (*p* < 0.05), whereas that in the S phase decreased significantly (*p* < 0.05) (Figure 9b). Overall, these results indicate that FGF10 overexpression affects cell cycle progression in SHF-DPCs, with interference showing effects opposite to overexpression.

Compared to the control group, the proportion of miR-184-overexpressing cells in the G2/M phase was not significantly different, that in the G0/G1 phase significantly increased (*p* < 0.05), and that in the S phase significantly decreased (*p* < 0.01) (Figure 9c). In the miR-184 knockdown group, the proportion of cells in the G0/G1 phase decreased significantly (*p* < 0.01), that in the S phase increased significantly (*p* < 0.05), and that in the G2/M phase increased significantly (*p* < 0.05) (Figure 9d). Overall, these results indicate that miR-184 overexpression can affect the cell cycle progression of SHF-DPCs, with interference having the opposite effects.

## 4. Discussion

The economic traits, such as yield and quality, of cashmere are influenced by factors such as HF growth and development as well as nutrient metabolism levels. Clarifying the regulatory mechanism of HF growth and development is crucial for improving cashmere quality and increasing its yield. Melatonin can inhibit the apoptosis of skin SHF cells through an antioxidant effect, promoting their development [24]. Xu et al. found a correlation between KRT26 and TCHH gene polymorphisms and cashmere fineness in Liaoning cashmere goats [25]. Wang et al. conducted a genome-wide association study on Inner Mongolia cashmere goats and found that FGF1, SEMA3D, EVPL, and SOX5 genes were related to cashmere yield, fineness, and length [26]. Wu et al. revealed the role of lncRNA in the fibre diameter of Chinese cashmere through omics analysis [27]. Liu et al. sequenced the transcriptome of skin tissues in the growing and resting stages and found that *FZD6*, *LEF1*, *FZD*, *WNT5A*, *TCF7*, and other differentially expressed genes in the Wnt signal pathway were regulated by miR-195, miR-148a, and miR-4206 [28]. Yang et al. found that KAPs and KRTAPs are closely related to the periodic growth of SHFs, suggesting that they are related to the growth of HFs and the cycle of cashmere [29]. In summary, elucidating the regulatory mechanism of HD growth and development is important for improving cashmere quality and increasing cashmere yield.

As accessory organs of the skin, HFs are crucial for mammals to resist harsh environments and offer protection. The induction and development of embryonic SHFs requires the interaction of multiple cells in the skin microenvironment along with spatiotemporal integration of various signalling molecules. The MAPK, PI3K-AKT, RAS, and ECM receptor pathways, along with other pathways, are crucial for the induction and formation of SHFs [30,31]. In the present study, the host genes of the DEmRNAs and DEncRNAs were also highly enriched in these pathways. The KRT and KRTAP families associated with vitellogenesis were significantly enriched in the *Staphylococcus aureus* infection and oestrogen signalling pathways. The activation of the MAPK signalling pathway is regulated by signalling factors and G-protein-coupled receptor complexes in the oestrogen signalling pathway [32]. Moreover, the significant enrichment of FGF family members in HF-related metabolic pathways has attracted attention. Based on the analysis of the differential FGF family and its receptor gene construction obtained from sequencing, this family was not only significantly enriched in HF-development-related pathways but also had regulatory potential in cancer-related pathways, stem cell pluripotency regulation, and actin cytoskeleton regulation. The FGF family, a class of cytokines, plays an important regulatory role in biological homeostasis and disease occurrence [33]. In recent years, studies have shown that subcutaneous injection of FGF18 in mice can promote the early entry of resting HFs into the growth phase, indicating its involvement in the regulation of the HF cycle [34]. FGF2, an angiogenic regulator, has been shown to induce angiogenesis and stimulate HF growth [35]. In addition, in vitro experiments and animal models have revealed a regulatory role for the FGF family in HF development [36,37].

Currently, ceRNA regulatory networks are used as a research tool to reveal biological phenomena. The miR-200 family is preferentially expressed in the epidermis and has regulatory effects on SHF cell proliferation and cell adhesion [38]. miR-24 affects HF morphogenesis through the targeted regulation of Tcf-3 [39]. The target genes of the let-7 family play important roles in the VEGF, TGF-β, and NF-κB signalling pathways [40]. The key miRNAs for HF development mentioned above were also identified in the network graph constructed in this study and were found to involve many ncRNA pairs. For example, the miR-200 family predicted a total of 135 ncRNA pairs. Therefore, we speculate that the miR-200 family transmits molecular signals located in the epidermis to HF cells through a rich ceRNA network, thereby regulating cellular processes. Zhang et al. [8] showed that lncRNA-H19 involves the miR-214-3p/β-catenin axis in promoting the proliferation of DPCs in cashmere goats. Moreover, circRNA3236 has a targeted binding relationship with miR-27b-3p and miR-16b-3p during HF development [41]. In the present study, we also discovered potential ncRNA relationship pairs centred on other miRNAs, such as miR-214-3p, EDAR, lncRNA000005916, and miR-432-3p, which require unified verification and discussion in subsequent experiments. In summary, the ceRNA network provides detailed association information for the screening of key HF development candidates, helping researchers gain a deeper understanding of the role of ncRNAs in SHF development and their underlying regulatory mechanisms.

FGF10 is a key molecule involved in HF development. FGF10-knockout mice show HF hypoplasia and dysplasia development in some organs [42]. The receptor-specific expression of FGF10 in epithelial cells promotes the proliferation and differentiation of HF stem cells, indicating that FGF10 is important in the development of various organs, including HFs, and for the repair of skin wounds. In addition, miR-184 is a highly conserved miRNA that plays important roles in cell growth, differentiation, and apoptosis. miR-184 plays a role as a tumour suppressor in glioblastomas and neuroblastomas [43,44]. CircRNA-ZNF609 acts as a miR-184 spike in corneal neovascularisation, blocking miR-184 activity and promoting the proliferation, migration, and angiogenesis of human corneal epithelial cells [45]. Moreover, miR-184 can participate in corneal vascular inhibition through the AKT/VEGF signalling pathway [46]. Nagosa et al. [47] found that miR-184 is highly expressed in the HF matrix, expressed at low levels in the epidermal matrix, and regulates the balance between epidermal proliferation and differentiation by inhibiting FIH1 and keratin K15, a marker of resting HF cells. These findings suggest the role of miR-184 in HF development, warranting further investigation.

The present study had some limitations. Although this study confirms that FGF10 affects the development of SHF-DPCs, it did not clarify the intermediate processes by which FGF10 affects the development of SHF-DPCs and SHFs in cashmere goats due to the lack of previous research studies. Moreover, this study mainly focused on the interactions between FGF10 and miR-184 and the mechanism of action on SHF-DPCs but lacks sufficient validation of the results related to circRNAs. This aspect will be the focus of our subsequent studies. In addition, in future studies, we will further investigate the underlying mechanisms of FGF10 in SHF development and further validate the role of chi-circRNA-0001141 in refining the ceRNA network by investigating the effect of the interaction between miR-184 with FGF10 on SHF-DPCs. We believe that these subsequent studies will help us better understand the roles of FGF10 and ncRNA during SHF development. Nevertheless, this study contributes to the advancement of our knowledge of the mechanisms of HF development and provides a valuable reference for future related studies.

## 5. Conclusions

In this study, whole-transcriptome sequencing was used to construct a ceRNA regulatory network for the developmental stages of cashmere goat HFs. Dual-luciferase reporter gene vector experiments revealed miR-184/FGF10 interactions. Subsequent experiments revealed that miR-184 overexpression significantly reduced the expression of FGF10, leading to the inhibition of SHF-DPC proliferation and an increase in SHF-DPC apoptosis. Cell cycle analysis showed that the proportion of cells in the G0/G1 phase increased and that in the S phase decreased. Moreover, our results suggest that miR-184 can release competitively adsorbed FGF10, resulting in elevated FGF10 expression, causing an increase in the proportion of SHF-DPCs in the S phase, promoting cell proliferation, inhibiting apoptosis, and affecting SHF development, which in turn affects the growth of cashmere (Figure 10).

## Figures and Tables

**Figure 1 animals-13-02234-f001:**
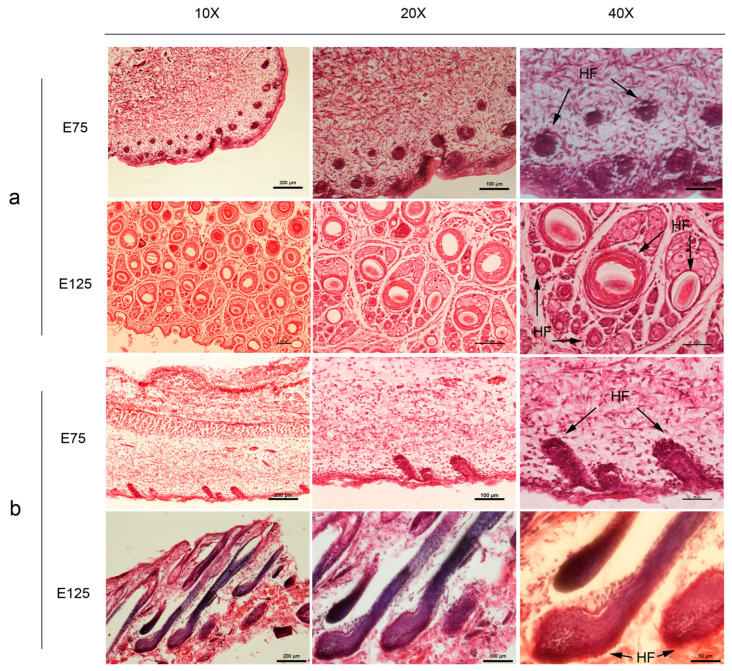
Haematoxylin and eosin staining of skin sections. (**a**) Transverse sections of E75 and E125 skin samples. (**b**) Longitudinal sections of E75 and E125 skin samples. HF: hair follicle. Scale bars: 10× scale, 200 μm; 20× scale, 100 μm; 40× scale, 50 μm.

**Figure 2 animals-13-02234-f002:**
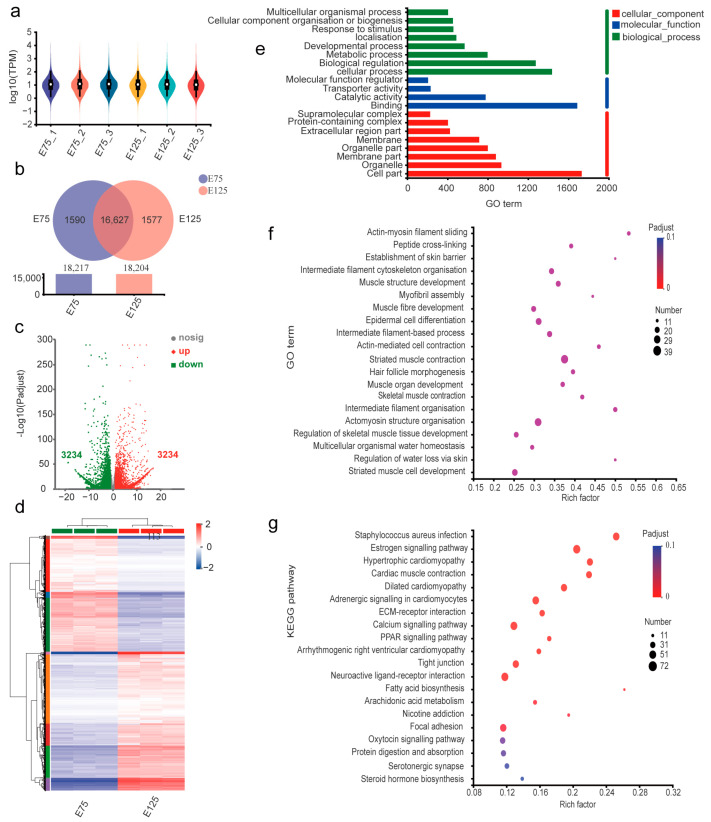
Identification and analysis of E75 and E125 differentially expressed (DE) messenger RNAs (mRNAs). (**a**) Violin plots of gene expression patterns for each sample. (**b**) Venn diagram of mRNA distribution across samples, where the overlap represents consensus mRNA for two sample types. (**c**) Volcano plots of log2-fold change (FC) for E75 and E125 mRNA. (**d**) Cluster analysis heatmap. (**e**) Gene Ontology (GO) functional annotations. (**f**,**g**) GO functional enrichment and Kyoto Encyclopedia of Genes and Genomes (KEGG) signalling pathway enrichment. Circle size and colour represent the degree of enrichment and enrichment significance, respectively.

**Figure 3 animals-13-02234-f003:**
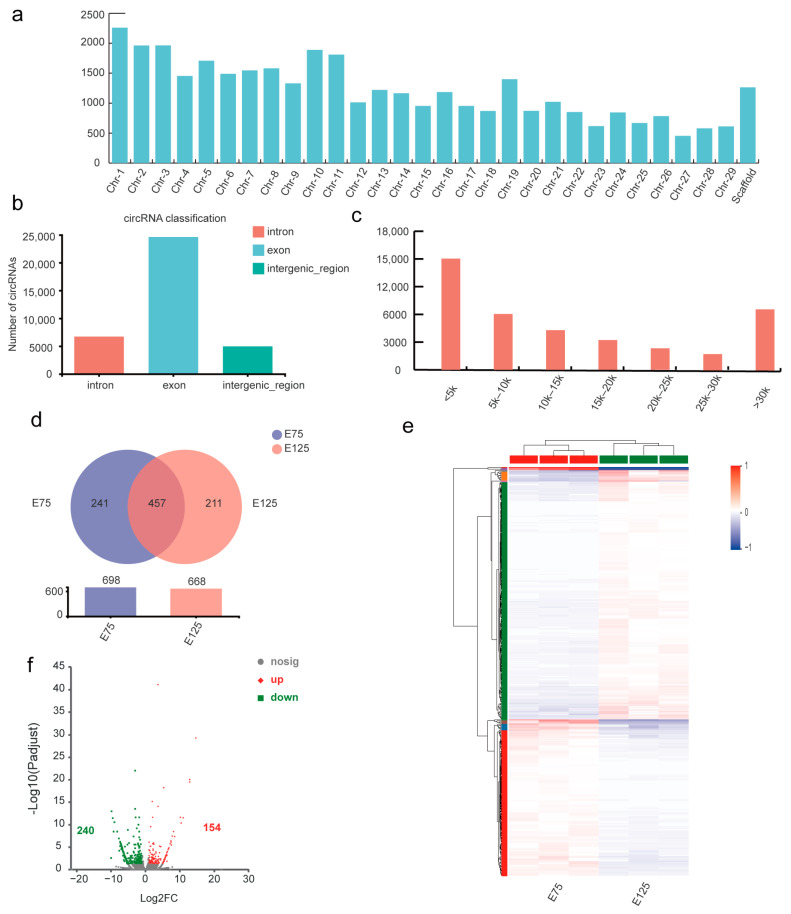
Identification and analysis of DEcircRNAs in E75 and E125. (**a**) CircRNA chromosomal distribution map. (**b**) CircRNA type distribution map, where exons are circRNAs with start and stop points located in the exon regions of genes, introns are circRNAs located at an intron region of a gene at the start or end point, and intergenic means that either one end of the start or end point is located on an intergene region or both start and end points are located in an exon region of a gene but the two exons are not in the circRNA of the same gene. (**c**) CircRNA length distribution plots with the ordinate as the number of circRNAs and the abscissa as the circRNA length distribution. (**d**) Venn plot of circRNA distribution across samples, where the overlap represents consensus circRNAs for the two sample types. (**e**) Cluster analysis heatmap. (**f**) Volcano plots of log2 FC (ST/CK) for E75 and E125 circRNAs.

**Figure 4 animals-13-02234-f004:**
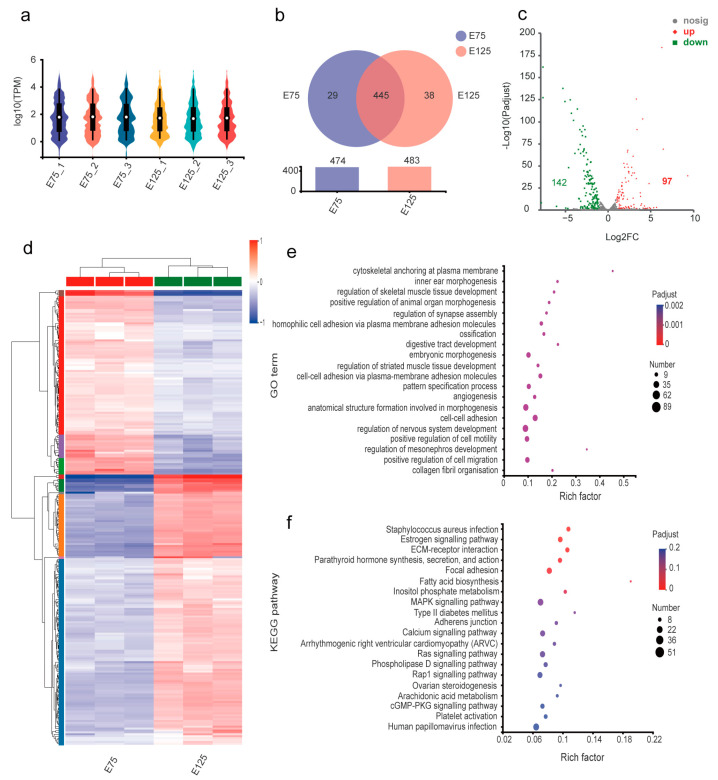
Identification and analysis of E75 and E125 DEmiRNAs. (**a**) Violin plots of gene expression patterns for each sample. (**b**) Venn diagram of miRNA distribution across samples with the intersecting section representing shared miRNAs of both samples. (**c**) Volcano plot of log2 FC (ST/CK) for E75 and E125 miRNAs. (**d**) Cluster analysis heatmap. (**e**,**f**) GO functional enrichment and Kyoto Encyclopedia of Genes and Genomes (KEGG) signalling pathway enrichment. The circle size and circle colour represent enrichment and enrichment significance, respectively.

**Figure 5 animals-13-02234-f005:**
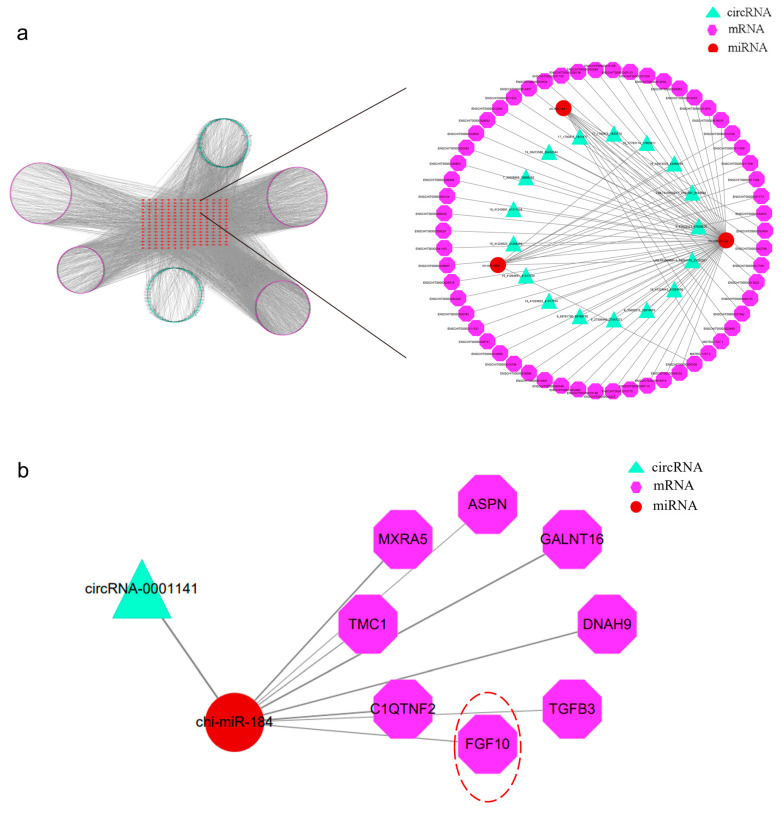
Diagram of the ceRNA regulatory network. (**a**) The ceRNA regulatory network composed of miR-184, miR-200b, and miR-214-3p. (**b**) The ncRNA axis of action constituted by chi-circRNA-0001141/chi-miR-184/FGF10.

**Figure 6 animals-13-02234-f006:**
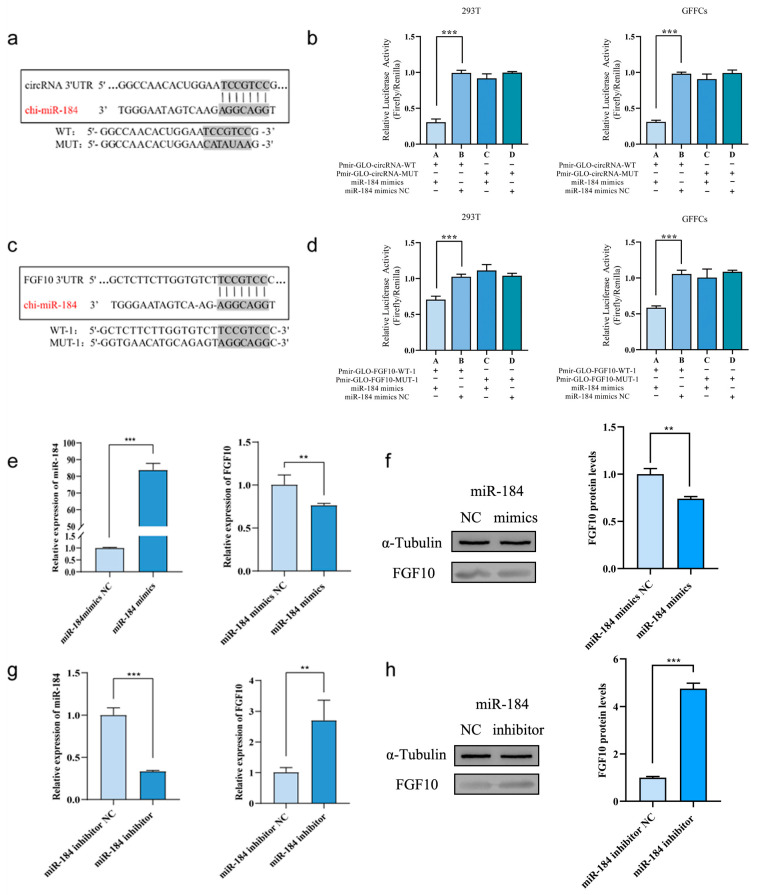
Relationship between FGF10 and ncRNA targeting. (**a**) Predicted site map for chi-circRNA-0001141 and miR-184. (**b**) Results for chi-circRNA-0001141 and miR-184 dual fluorescence assay (miRanda prediction). (**c**) Predicted site map for miR-184 and *FGF10* (miRanda). (**d**) Results of miR-184 and *FGF10* dual fluorescence assay (miRanda prediction). (**e**) Relative expression of miR-184 and *FGF10* after mimic treatment. (**f**) FGF10 expression after miR-184 overexpression, with grey-scale analysis of expression levels. (**g**) Relative expression of miR-184 and *FGF10* after inhibitor treatment. (**h**) FGF10 protein expression after miR-184 interference, with greyscale analysis of expression levels (** *p* < 0.01, *** *p* < 0.001).

**Figure 7 animals-13-02234-f007:**
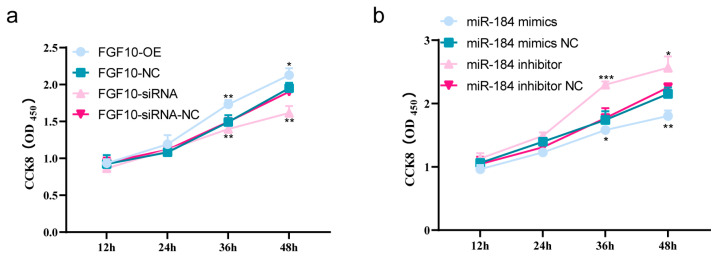
miR-184 inhibition of proliferation of SHF-DPCs by targeting of FGF10. (**a**) Effect of FGF10 on cell proliferation. (**b**) Effect of miR-184 on cell proliferation (* *p* < 0.05, ** *p* < 0.01, *** *p* < 0.001).

**Figure 8 animals-13-02234-f008:**
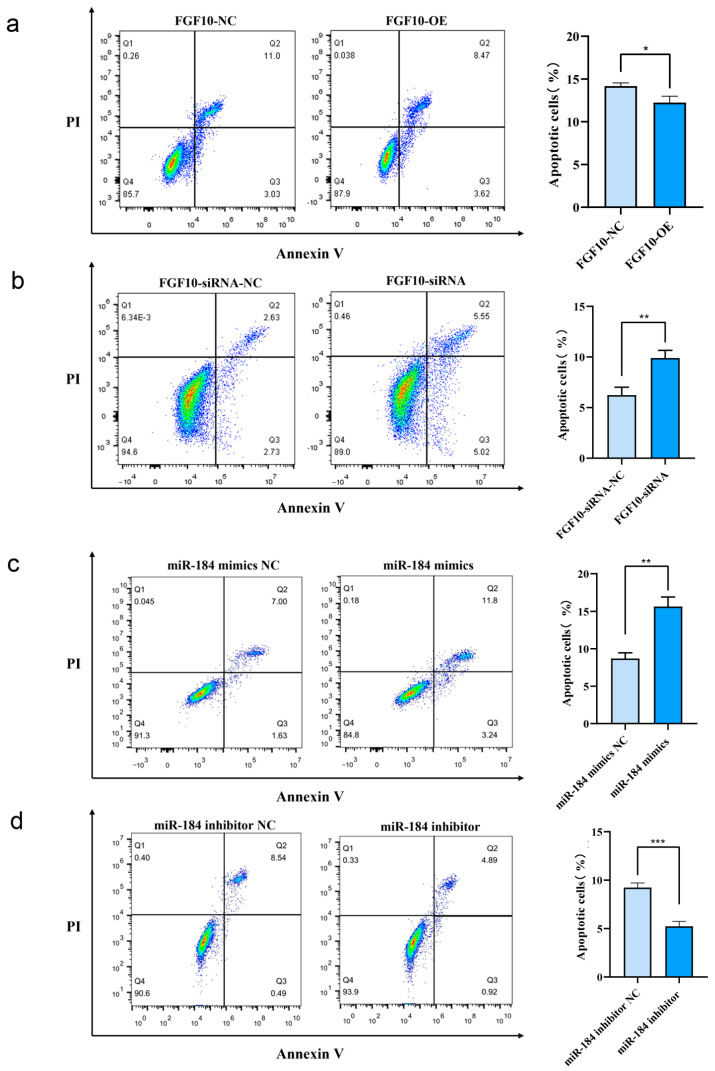
miR-184 inhibition of apoptosis in SHF-DPCs by targeting of FGF10. (**a**) Effect of *FGF10* overexpression on apoptosis, with the percentage of apoptotic cells. (**b**) Effect of *FGF10* interference. (**c**) Effect of miR-184 overexpression. (**d**) Effect of miR-184 inhibition (* *p* < 0.05, ** *p* < 0.01, *** *p* < 0.001).

**Figure 9 animals-13-02234-f009:**
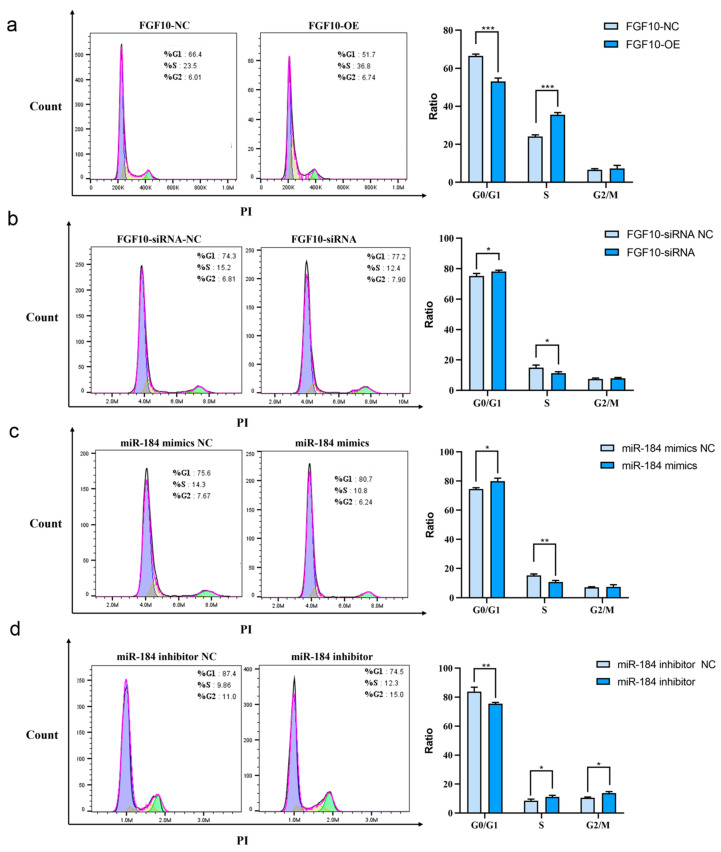
Effect of miR-184 on cell cycle progression in SHF-DPCs by targeting of FGF10. (**a**) Effect of *FGF10* overexpression on the cell cycle, with cell numbers by phase. (**b**) Effect of FGF10 interference. (**c**) Effect of miR-184 overexpression. (**d**) Effect of miR-184 interference (* *p* < 0.05, ** *p* < 0.01, *** *p* < 0.001).

**Figure 10 animals-13-02234-f010:**
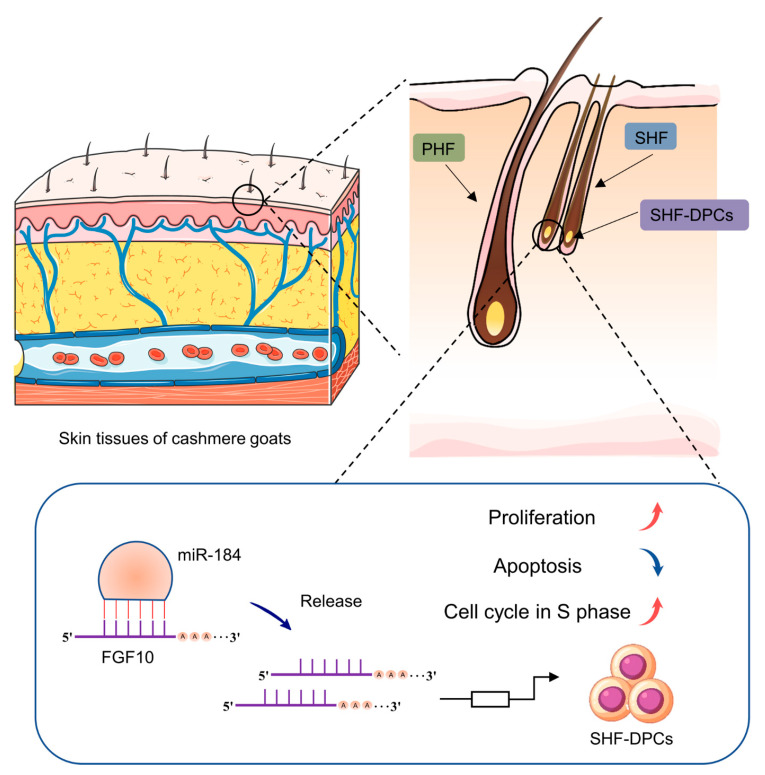
Model for regulation of SHF-DPCs by FGF10-related ceRNA.

## Data Availability

All data are presented in the manuscript.

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
