# Peer review of "Effect of Fibroblast Growth Factor 10 and an Interacting Non-Coding RNA on Secondary Hair Follicle Dermal Papilla Cells in Cashmere Goats’ Follicle Development Assessed by Whole-Transcriptome Sequencing Technology"

_animals, 2023, doi:10.3390/ani13132234_

Round 1

Reviewer 1 Report

The paper presents that this study systematically investigated the network of ceRNA actions during hair follicle development in the cashmere goat using whole transcriptome techniques. Further discoveries,FGF10 and the interacting miR-184 play important roles in dermal papilla cells of cashmere goat secondary hair follicles. This has important implications for understanding the developmental mechanisms of the hair follicle in the cashmere goat, it is a topic of interest to researchers in related areas. However, a few major issues need to be addressed. Therefore, we believe that your paper requires revisions at this point.

Main Evaluations and Issues:

1.         Why were E75 and E125 chosen for whole-transcriptome sequencing and are these two periods representative of hair follicle development?

2.         I noticed several instances where the descriptions were unclear, for example: full transcriptome in line 16 and Whole-transcriptome in line 69. Please review the manuscript carefully and make the necessary corrections to ensure that your study's presentation is clear and accurate.

3.         Please use Oxford commas and use a format like A, B, and C when listing to avoid ambiguity. For example: lines 82 and 83.

4.         In the discussion section, the manuscript does not provide any direction on future research or the next steps following this study's completion. Please consider providing some guidance in this area.

5.         It would be helpful if you discussed the limitations of your study more explicitly.

6.         The description of the conclusions does not appear to be fully consistent with Figure 11, please correct this. Although the conclusions drawn are adequately supported by the data, we believe they could be presented in a more appealing and clear manner to better convey their significance and impact.

7.         It remains unclear what the potential value of your study is. We suggest that you articulate this aspect more clearly and persuasively in the manuscript.

It is recommended to make minor changes to the quality of English, especially in terms of grammar and terminology.

Reviewer 2 Report

This study investigated the ncRNA and mRNA expression and constructed the ceRNA regulatory network on secondary hair follicle dermal papilla cells in Cashmere goats. The anthors validated the targeted binding relationship of circRNA-0001141, miR-184, FGF10, and also discovered that miR-184 affects dermal papilla cell proliferation and apoptosis by regulating FGF10 expression. I think this study can provide some basis for subsequent exploring the mechanism of the development of cashmere hair follicles. I recommend some recision, with the following comments.

1. Language editing of this manuscript must be performed by professionals. Some phrases in the manuscript are unclear.

2. The authors selected ewes with similar cashmere production performance and used their lamb for sequencing analysis. I was curious as to whether cashmere production performance could be consistently inherited. In addition, why choose E75 and E125, there is no consideration of other time points, such as after birth, etc.

3. Some of the software and methods involved in the analysis of biological information should be added to the references or online sites, such as CPC, CNCI, CIRI2.

4. Primer sequences for RT-qPCR should be provided Gene ID and product length.

5. The result of lncRNA was put in the manuscript, but the subsequent content did not involve the related content of lncRNA, it is suggested to delete the related content.

6. Validation of circRNA was only carried out by RT-qPCR. Additional experiments are needed to verify the circular structure of circRNA, and whether the splice sites were taken into account when designing RT-qPCR primers for circRNA. The authors need to name the circRNAs according to the nomenclature.

7. The authors verified the target binding relationship of circRNA-0001141-miR-184-FGF10, and just carried out the functional validation of miRNA regulation of FGF10. I think that the experimental validation was too simple and incomplete, and it is suggested to perform the functional validation of circRNA. Otherwise there is no way to illustrate the final conclusion of the ceRNA network in this manuscript.

8. The article also has some writing problems, for example Line 31  obtained 6468 shoud be obtained 6,468, it needs further revision.

Language editing of this manuscript must be performed by professionals. Some phrases in the manuscript are unclear.

Reviewer 3 Report

Manuscript seems long and figures and tables were many. But sorry, I could not know what was the novel finding from this experiment.

The paragraph for NGS (L63-L77) seems redundant for me. Did this paragraph explain a motivation/hypothesis/proposition by the authors?

The authors should explain the feeding/rearing conditions of pregnant ewes used. Furthermore, why did the authors used foetal skin samples? There seems not explanation for this point. Moreover, I could not find the number of samples used in Materials and Methods section. Where the foetal skins were sampled from?

The authors should explain the setting for NGS analysis in more detail (e.g., read depth).

Did Figure 11 show the mechanism at foetal stage?

There seems sentences which were hard to follow around the manuscript (e.g., L93-L95). Please check carefully.

Reviewer 4 Report

Introduction:

Some general comments on cashmere wool are desired to explain seasonality of wool growth, enriched pathways in species with fine cashmere and RNA-analyses in cashemere goats.

Please add previous reports on cashmere:

Transcriptomes reveal microRNAs and mRNAs in different photoperiods influencing cashmere growth in goat.

Liu B, Zhao R, Wu T, Ma Y, Gao Y, Wu Y, Hao B, Yin J, Li Y. PLoS One. 2023 Mar 17;18(3):e0282772. doi: 10.1371/journal.pone.0282772. eCollection 2023.

Convergent Genomic Signatures of Cashmere Traits: Evidence for Natural and Artificial Selection.

Wang W, Li Z, Xie G, Li X, Wu Z, Li M, Liu A, Xiong Y, Wang Y. Int J Mol Sci. 2023 Jan 6;24(2):1165. doi: 10.3390/ijms24021165.    

Association between the cashmere production performance, milk production performance, and body size traits and polymorphism of COL6A5 and LOC102181374 genes in Liaoning cashmere goats. Zhang Y, Qin Y, Gu M, Xu Y, Dou X, Han D, Lin G, Wang L, Wang Z, Wang J, Sun Y, Wu Y, Chen R, Qiao Y, Zhang Q, Li Q, Wang X, Xu Z, Cong Y, Chen J, Wang Z. Anim Biotechnol. 2022 Dec 17:1-15. doi: 10.1080/10495398.2022.2155177    

Identification of the key proteins associated with different hair types in sheep and goats. Zhang C, Qin Q, Liu Z, Xu X, Lan M, Xie Y, Wang Z, Li J, Liu Z. Front Genet. 2022 Sep 23;13:993192. doi: 10.3389/fgene.2022.993192. eCollection 2022.      

Expression profile of long non-coding RNA in inner Mongolian cashmere goat with putative roles in hair follicles development. Ma R, Shang F, Rong Y, Pan J, Wang M, Niu S, Qi Y, Li Y, Lv Q, Wang Z, Wang R, Su R, Liu Z, Zhao Y, Wang Z, Li J, Zhang Y. Front Vet Sci. 2022 Sep 2;9:995604. doi: 10.3389/fvets.2022.995604. eCollection 2022.      

Effects of lncRNA MTC on protein expression in skin fibroblasts of Liaoning Cashmere goat based on iTRAQ technique. Jin M, Fan W, Piao J, Zhao F, Piao J. Anim Biotechnol. 2022 Sep 10:1-10. doi: 10.1080/10495398.2022.2119406    

Comprehensive Transcriptome Analysis of Hair Follicle Morphogenesis Reveals That lncRNA-H19 Promotes Dermal Papilla Cell Proliferation through the Chi-miR-214-3p/β-Catenin Axis in Cashmere Goats. Zhang Y, Li F, Shi Y, Zhang T, Wang X. Int J Mol Sci. 2022 Sep 2;23(17):10006. doi: 10.3390/ijms231710006.    

The regulation mechanism of different hair types in inner Mongolia cashmere goat based on PI3K-AKT pathway and FGF21. Gong G, Fan Y, Zhang Y, Yan X, Li W, Yan X, He L, Wang N, Chen O, He D, Jiang W, Li J, Wang Z, Lv Q, Su R. J Anim Sci. 2022 Nov 1;100(11):skac292. doi: 10.1093/jas/skac292.  

Sequence Variation in Caprine KRTAP6-2 Affects Cashmere Fiber Diameter. Cao J, Wang J, Zhou H, Hu J, Liu X, Li S, Luo Y, Hickford JGH. Animals (Basel). 2022 Aug 11;12(16):2040. doi: 10.3390/ani12162040.    

Identification of Genes Related to Hair Follicle Cycle Development in Inner Mongolia Cashmere Goat by WGCNA. Gong G, Fan Y, Yan X, Li W, Yan X, Liu H, Zhang L, Su Y, Zhang J, Jiang W, Liu Z, Wang Z, Wang R, Zhang Y, Lv Q, Li J, Su R. Front Vet Sci. 2022 Jun 14;9:894380. doi: 10.3389/fvets.2022.894380. eCollection 2022.  

Identification of the Key Genes Associated with Different Hair Types in the Inner Mongolia Cashmere Goat. Gong G, Fan Y, Li W, Yan X, Yan X, Zhang L, Wang N, Chen O, Zhang Y, Wang R, Liu Z, Jiang W, Li J, Wang Z, Lv Q, Su R. Animals (Basel). 2022 Jun 4;12(11):1456. doi: 10.3390/ani12111456.  

Screening of microRNA and mRNA related to secondary hair follicle morphogenesis and development and functional analysis in cashmere goats. Shang F, Wang Y, Ma R, Rong Y, Wang M, Wu Z, Hai E, Pan J, Liang L, Wang Z, Wang R, Su R, Liu Z, Zhao Y, Wang Z, Li J, Zhang Y. Funct Integr Genomics. 2022 Oct;22(5):835-848. doi: 10.1007/s10142-022-00842-y.  

There might even be more previous reports on cashmere and its regulation.      

Materials and Methods

Please give the number of ewes and embryos used fort his study.

Can you give data on cashmere quality of parents and the reason for selection animals of this line.

Can you state from which tissues samples were taken and which samples were used for RNASeq.

Please refer to previous work in the discussion. Also you may discuss if the type/location of tissue and the season when the experiment was performed  may have had an influence on the outcome of the present study.

No comments

Round 2

Reviewer 3 Report

The manuscript has been greatly improved.

I would like to propose the authors to add some description about the response, or the authors' belief, to my previous comment No. 5 ("Did Figure 11 show the mechanism at foetal stage?").
